# Quantitative Proteomic Characterization of Foreign Body Response towards Silicone Breast Implants Identifies Chronological Disease-Relevant Biomarker Dynamics

**DOI:** 10.3390/biom13020305

**Published:** 2023-02-06

**Authors:** Ines Schoberleitner, Klaus Faserl, Bettina Sarg, Daniel Egle, Christine Brunner, Dolores Wolfram

**Affiliations:** 1Department of Plastic, Reconstructive and Aesthetic Surgery, Medical University of Innsbruck, Anichstraße 35, A-6020 Innsbruck, Austria; 2Protein Core Facility, Biocenter, Institute of Medical Chemistry, Medical University of Innsbruck, Innrain 80-82, A-6020 Innsbruck, Austria; 3Department of Obstetrics and Gynecology, Medical University of Innsbruck, Anichstraße 35, A-6020 Innsbruck, Austria

**Keywords:** SMI (silicone mammary implants), FBR (foreign body response), wound healing, wound infection, capsular fibrosis, implant encapsulation, early-stage fibrosis, surface adsorption, immunomics, biomarkers

## Abstract

The etiology of exaggerated fibrous capsule formation around silicone mammary implants (SMI) is multifactorial but primarily induced by immune mechanisms towards the foreign material silicone. The aim of this work was to understand the disease progression from implant insertion and immediate tissue damage response reflected in (a) the acute wound proteome and (b) the adsorption of chronic inflammatory wound proteins at implant surfaces. An intraindividual relative quantitation TMT-liquid chromatography–tandem mass spectrometry approach was applied to the profile wound proteome formed around SMI in the first five days post-implantation. Compared to plasma, the acute wound profile resembled a more complex composition comprising plasma-derived and locally differentially expressed proteins (DEPs). DEPs were subjected to a functional enrichment analysis, which revealed the dysregulation of signaling pathways mainly involved in immediate inflammation response and ECM turnover. Moreover, we found time-course variations in protein enrichment immediately post-implantation, which were adsorbed to SMI surfaces after 6–8 months. Characterization of the expander-adhesive proteome by a label-free approach uncovered a long-term adsorbed acute wound and the fibrosis-associated proteome. Our findings propose a wound biomarker panel for the early detection and diagnosis of excessive fibrosis that could potentially broaden insights into the characteristics of fibrotic implant encapsulation.

## 1. Introduction

Fibrosis is a hallmark of numerous pathological conditions and is marked by the excessive production and accumulation of collagenous and non-collagenous extracellular matrix (ECM) components, leading to a loss of tissue architecture and organ function [1]. Initiated by injury and consecutive inflammation (e.g., persistent infections, autoimmune reactions, allergic responses, chemical insults, radiation, and tissue injury), the subsequent continuum of complex and sequential chronic inflammatory reactions tunnels into TGFβ, Smad, NF-κB, and MAPK signaling cascades [2,3,4,5,6,7,8,9,10,11,12]. 

Silicone is the most widely used implant material in routine medical practice, despite side effects, such as the formation of capsular contracture causing pain, a distinguishing aesthetic result, and the impairment of implant function [13,14,15,16]. No matter how noninvasive, the surgical implantation of foreign material causes injury that can initiate the fibrotic response [17,18]. 

Over the last 15 years, we have intensively investigated silicone breast implant-induced types of fibrosis with the objective of fundamental and clinical research [11,19,20,21,22,23]. After any injury, the body suffers inflammation, matrix formation, and fiber rearrangement [11]; however, if the infection is associated with the biomaterial, the degree of fibrosis will dramatically increase [18]. We consider silicone mammary implants (SMIs) a paradigmatic example of foreign body-induced fibrotic diseases [11,21,22], and our findings are also relevant for fibrotic complications in carriers of other active (cardiac pacemakers, insulin pumps, etc.) and passive (drainages, catheters, medical tubing, etc.) silicone-based devices [13,21,24,25,26,27,28]. Our generated data have clearly shown that the earliest stage of fibrosis always includes an inflammatory phase, which is characterized by the cells of the innate and adaptive immune system [11,19,21,22,23].

Generally, fibrosis is perceived as a reparative or reactive process [11], mainly due to the proliferation and activation of fibroblasts and myofibroblasts. Innate immune cells, such as neutrophils (during bacterial infections), eosinophils (during parasitic infections), monocytes/macrophages, mast cells, and natural killer/natural killer T (NK/NKT) cells, are found in the injured tissue. Infections or injuries resulting in inflammation initiate minor shifts in tissue homeostasis, leading to the migration and accumulation of inflammatory cells in the profibrotic environment inducing profibrotic signaling molecules and accelerating the development of fibrosis [11,29] per se. Among these cells, macrophages, neutrophils, and mast cells are considered to be the major players in innate immune response, i.e., phagocytosis of tissue debris, dead cells, and any foreign organisms or materials, leading to fibrosis [30]. Upon macrophage activation, T cells migrate to the injury site, infiltrate the connective tissue, and release profibrotic cytokines and inflammatory mediators that stimulate/activate fibroblasts (and other cells) to transdifferentiate into α-smooth muscle actin (SMA)-expressing myofibroblasts [31,32,33]. If the activation of myofibroblasts persists, due to insufficient elimination of the initial inflammatory stimulus (e.g., damaged tissue, foreign organisms, or foreign materials), an overproduction of ECM proteins would lead to the formation of collagen-I-rich (III > I) fibrous tissue, and ECM-degrading enzymes (such as metalloproteinases, MMPs) influence subsequent wound healing and fibrotic responses by leaving an irreversible scar that impairs the function of the organ or tissue [34,35]. 

Medical device performance and biocompatibility are both directly related to unwanted side effects such as foreign body response, inflammation, and cell adhesion [26]. Silicone is a sticky and viscous material and can bind non-specifically to blood proteins [36]. When implants are inserted and come into contact with the leakage of blood at the wound site, blood biomaterial interactions emphasize protein absorption to the surface of biomaterials [37]. The adverse reaction to the adhesive proteome reflects in the activation of coagulation and leukocytes that produce inflammation, adhesion, and the activation of platelets [37,38]. Moreover, inflammatory reactions to silicone elastomer particulate debris [39]—specifically particle shedding from breast implants [40]—have been demonstrated. In particular, macrophages and foreign body giant cells (FBGCs) take up silicone [41,42,43]. Together with neutrophils, mast cells, and fibroblasts, these cells have all been implicated in sensing the biomaterial and in producing signals that may alter fibrotic responses [25,41]. 

Protein adsorption to the surface of silicon mammary implants (SMI) has been investigated, either by incubation with serum proteins [19,20,44,45], or post-operatively, by stripping off the proteome after implant removal. However, no studies have been performed under controlled clinical conditions using intraindividual periodical sampling in breast cancer patients, from the acute wound, in the first five days and during early-stage fibrosis approx. 8 months post-implantation.

Of note, the serum is not the only origin of foreign body response towards implants. Tissue injury after the surgical insertion of SMI immediately activates the innate immune system, setting in motion a local inflammatory response and proinflammatory mediation that includes the recruitment of inflammatory cells from the circulation [11,29].

Now, wound fluid is commonly used for protein profiling and analysis. However, the correct method of sample collection is crucial in highly sensitive proteomic analyses [46,47]. For untargeted proteomics and biomarker discovery studies, the simultaneous identification and measurement of large protein numbers using mass spectrometry is favored, yet rare in implant-immunoreactivity research [48]. 

To date, there is no existing study of SMI-adsorbed immunoreactive proteome, from the acute wound after tissue injury to implant encapsulation and early-stage fibrosis, due to chronic inflammation. Based on the idea that apart from silicone particulate shedding, wound protein surface adsorption represents a long-term inflammatory trigger, markers alone from serum post-op may not aid in the precise molecular diagnosis or even as a predictor of performance. Therefore, we sought to identify comprehensive and distinctive marker profiles of the FBR against SMI in the acute wound milieu and later, associated with the implant surface 6–8 months after surgery. In this context, we applied a functional proteomic approach to the analysis of plasma (pre-op) as well as wound bed fluid (1–5 days post-op), and tissue expander surface-adsorbed and adhered proteome (6–8 months post-op) from breast cancer patients undergoing simultaneously prophylactic NSME and tissue expander (inflatable implant)-based breast reconstruction. To achieve the highest possible biological significance, we intraoperatively compared two tissue expanders with varying topography: (i) the conventionally used tissue expander at our Department—the CPX^®^ 4 (roughness radius: 60 µM of Ra; Mentor) and (ii) the novel, surface-roughness-reduced device SmoothSilk^®^ (roughness: 4 µM of Ra; Motiva) to determine the common global SMI-adhesive proteome, adhered to both expanders in all tested subjects, albeit independent of silicon chemistry [49] or surface roughness [22,50]. To track the dynamics of the protein expression of immediate inflammatory response and wound-healing mechanisms in the acute wound post-op, we used a Tandem Mass Tag TMT-based quantitative proteomic approach. To identify potential biomarkers for the early detection of early-stage fibrosis, we stripped the device-associated proteome from the surface after the tissue expanders had been exchanged with definite implants in a second operation, 6–8 months post-implantation. The set was compared with the acute wound proteome and analyzed for candidates that were associated long-term with the device surface. These results are the first to provide relevant information and insight into the aberrant progression of wound healing into fibrosis in real-time, in vivo. Moreover, we provide relevant information contributing to the discovery of novel late-stage candidate biomarkers of the disease and potential therapeutic targets, providing a foundation that helps to understand the causative relation between silicone mammary implants and the autoimmune response of the immune system.

## 2. Materials and Methods

### 2.1. Study Population

This study included a total of 7 female patients who were undergoing simultaneous prophylactic bilateral nipple-sparing mastectomy (NSME) and tissue expander-based breast reconstruction. Informed consent for photo documentation, the operation, sample collection, and the anonymized evaluation and publication of data was obtained in written form from all patients after confirmation of all inclusion and exclusion criteria (Table 1). 

Patient demographics including age; body mass index (BMI); and breast cup size; breast symmetry; previous scars in the breast area; comorbidities (chronic diseases, allergies, therapeutics); dominant hand; smoking habits; profession (manual labor/office job); and physical training habits, as well as duration, were documented. All donor biological samples (blood, wound bed fluid, and removed tissue expander) and information were obtained with the informed written consent of the participants, and by: (i) the regulations of the relevant clinical research ethics committee, as well as (ii) the Declaration of Helsinki, and with (iii) The European Union Medical Device Regulation (§ 40 Section 3 Medical Devices Act). Therefore, 70 WBF samples (7 patients × 5 days post-op × 2 tissue expander types) and 14 tissue expander surface strips (7 patients × 2 expanders) were evaluated.

### 2.2. Study Design

This monocentric, randomized, double-blind controlled clinical study was approved by the Institutional Ethical Committee of the Medical University Innsbruck, Austria (protocol code 1325/2019, 23 January 2020), and the Austrian Federal Office for Safety in Health Care (approval number; 13340962). Ten female patients undergoing bilateral prophylactic mastectomy and simultaneous tissue expander-based breast reconstruction, due to high-risk hereditary predisposition and/or confirmed *Brca1+*/*Brca2+* gene mutation, were enrolled in the study (Figure 1a).

To avoid the detection of a device-exclusive immune reaction and to analyze a general SMI-associated host response, we chose to implant two tissue expanders, both composed of a poly(dimethyl siloxane) (“PDMS”) elastomer shell, with varying surface topographies. All patients received both types of breast tissue expanders, the routinely used CPX^®^4 breast expanders (MENTOR^®^, Tysons, VA, USA: surface roughness: ~60 µM of Ra; from here referred to as *SMIb*), and the novel surface-roughness reduced SmoothSilk^®^ breast expanders (Motiva Flora^®^, Establishment Labs, Costa Rica: surface roughness: ~4 µM of Ra; from here referred to as *SMIa*), which were randomized to the left or right breast after bilateral prophylactic NSME (Figure 1b,c). One patient resigned and two patients were excluded from the study due to post-op complications. The patient and laboratory expert were double-blinded. Matching was performed intra-individually and conducted according to the implanted tissue expander. The inflatable tissue expanders were exchanged for definite implants in a second surgery, 6 to 8 months post-implantation.

### 2.3. Biological Sample Collection

A biological sample collection of wound bed fluid was performed daily from day 1 to 5 after expander implantation. The blood draw was performed along with anesthesia. Wound drains from patients undergoing expander-based reconstruction were left in situ post-operatively as part of the surgical procedure, and wound bed fluid (termed WBF) was collected under sterile conditions in sterile flasks on RT. For the first 24 h, no vacuum was applied to the drains; however, immediately after, the drains were left in situ with a vacuum until removal. Flasks containing WBF were removed every 24 h for the duration of 24–120 h (drains after total collection time; 120 h) post-operatively, and WBF was transported to the cell culture in our research laboratory. To obtain proteinaceous WBF, we gradient-separated the drain fluid by stratification on Ficoll-Paque^®^ (Cytivia) to remove the cellular fraction. The WBF was then sterilized by passing it through a 0.1 µM and subsequently 0.07 µm syringe filter to remove all cells, human and microbial. The proteinaceous fraction was frozen at −80 °C for further processing. 

Expander exchange with definite implants was performed during re-operation between 6 to 8 months after the initial expander implantation. Removed tissue expanders were placed immediately after withdrawal into sterile boxes and frozen, as well as stored at −80 °C before transport to the research laboratory.

### 2.4. Sample Preparation of Wound Bed Fluid and Serum Samples for TMT-Based Quantitative Proteomic Approach

#### 2.4.1. Immunoaffinity Depletion of Highly Abundant Plasma Proteins

The wound bed fluid and serum samples were subjected to immunoaffinity depletion of 14 highly abundant plasma proteins using the Multiple Affinity Removal Spin Cartridge, MARS Hu-14 (P/N: 5188–6560, Agilent Technologies, Santa Clara, CA, USA). Depletion was performed according to the manufacturer’s instructions. The starting material was a 30 µL sample that was diluted to 600 µL with Buffer A (P/N: 5185–5987, Agilent Technologies) and processed in three successive runs on the cartridge, using 200 µL each. 

#### 2.4.2. Protein Reduction and Alkylation

The proteins in 500 µL of flow-through were reduced by adding 50 µL of 100 mM dithiothreitol in Hepes buffer (100 mM of Hepes, pH8.5, 10 mM of CaCl_2_), followed by incubation at 56 °C for 30 min. Free cysteines were alkylated by adding 50 µL of 550 mM iodoacetamide in Hepes buffer, followed by agitation at room temperature for 20 min in the dark.

#### 2.4.3. Chloroform/Methanol Precipitation

Proteins were buffer-exchanged by chloroform/methanol precipitation [51] In brief, a 600 µL depleted sample was added to 600 µL of methanol and 150 µL of chloroform, vortexed, and centrifuged at 15,000 g for 10 min. The upper phase was discarded without disturbing the protein precipitate between the phases. Another 600 µL of methanol was added, vortexed, and the mixture centrifuged as before. The supernatant was discarded and the proteins in the pellet were dissolved in 95 µL of Hepes buffer. Total protein concentrations were determined according to Bradford (ROTI-Nanoquant assay, P/N: K880.3, Carl Roth GmbH).

#### 2.4.4. In-Solution Protein Digestion and TMT-Labeling

The samples were digested with 1 µg of trypsin (Sequencing Grade Modified Trypsin, P/N: V5111, Promega, Madison, WI, USA) overnight at 37 °C under agitation. Thereafter, one-sixth of every sample was withdrawn and pooled to obtain an internal reference sample. TMT 6-plex labeling was performed according to the manufacturer’s instructions. In brief, the samples were lyophilized to 25 µL and labeled individually with 10.3 µL of TMT 6-plex labeling. The reaction was incubated for 1 h at room temperature, quenched with 55 µL of ammonium-bicarbonate buffer (100 mM, pH 8.0) for 10 min, and acidified with 10 µL of 10% formic acid. The samples were combined to obtain the internal reference sample; all 5 wound bed fluid samples were collected at different time points from one patient and one implant side was present in one pooled sample. Serum samples from different patients were pooled together with an internal reference sample. Pools were lyophilized to 30 µL and subjected to high pH fractionation using the Pierce High pH Reversed-Phase Peptide Fractionation Kit (P/N: 84868).

### 2.5. Sample Preparation for Label-Free Quantitative Proteomic Analysis of Adsorbed Proteins to Tissue Expander Surface

#### 2.5.1. Excision of Tissue Expander Slices

To analyze proteins attached to the surface of the tissue expander, 1 cm^2^ slices were excised from frozen expanders using a scalpel. The silicon slices were transferred to 500 µL low-binding tubes with the surface directed inwards. Expander slices were washed with 200 µL of PBS for 10 min by agitation at room temperature. The supernatant was removed, and the slices were subjected to protein digestion. Three technical replicates were performed per tissue expander. 

#### 2.5.2. Protein Reduction, Digestion, and Alkylation

The proteins on the silicone slices were reduced by adding 200 µL of 10 mM dithiothreitol in ABC buffer (100 mM of ammonium-bicarbonate, pH 8.0), followed by incubation at room temperature for 1 h. For protein digestion, 2 µg of trypsin was added, and the reaction was agitated overnight at 37 °C. The alkylation of free cysteines was performed by adding 20 µL of 550 mM iodoacetamide in ABC buffer for 20 min in the dark. The supernatants were transferred to clean low-binding tubes, followed by desalting with C18 Tips (Pierce C18 Spin Tips, P/N: 87784, Thermo). The desalted peptides were dried and stored at −20 °C.

### 2.6. Liquid Chromatography Coupled to Tandem Mass Spectrometry (nanoLC-MS/MS)

Peptide digests were analyzed using an UltiMate 3000 nano-HPLC system coupled to a Q Exactive Plus mass spectrometer (Thermo Scientific, Waltham, MA, USA) equipped with a Nanospray Flex ionization source, as described previously (2). In brief, peptides were separated on a homemade column (100 μm i.d. × 17 cm length) packed with 2.4 μm of C18 material (ReproSil, Dr. A. Maisch HPLC GmbH). The solvents for nano-HPLC were 0.1% formic acid (solvent A) and 0.1% formic acid in 85% acetonitrile (solvent B). The total gradient time was 82 min at a flow rate of 300 nL/min. The 20 most abundant peptides in the survey scan were selected for MS fragmentation. The isolation window was set to 1.6 *m*/*z*. Survey full-scan MS spectra were acquired from 300 to 1750 *m*/*z* at a resolution of 60,000. Peptides were fragmented by HCD with normalized collision energy set to 28 for tissue expander samples and 32 for TMT-labeled wound bed fluid samples, respectively.

### 2.7. Database Search

The MS data files were processed using Proteome Discoverer version 2.2 (Thermo Scientific). MS/MS spectra were searched by the Sequest HT engine against a UniProt human reference proteome database (last modified 2 February 2022). The search parameters were as follows: Enzyme specificity was set to trypsin with two missed cleavages being allowed. Fixed modification was carbamidomethyl on cysteine; variable modifications were the oxidation of methionine and acetylation and/or methionine loss of the protein N-terminus. Precursor mass tolerance was set to 10 ppm; fragment mass tolerance was 20 mmu. The maximum false discovery rate (FDR) for protein and peptide identification was set to 1%. For label-free quantification of tissue expander samples, the Minora Feature Detector node was set to high-confidence PSM (peptide spectrum matches) only, with at least two isotopic peaks present in the isotope pattern. Retention time alignment was performed at a maximum retention time shift of 10 min and a mass tolerance of 10 ppm. For quantitation of the TMT-labeled wound bed fluid samples, the protein fold changes were calculated based on TMT reporter ion intensities present in MS2 scans (*m*/*z* 126, 127, 128, 129, 130, and 131). The reporter ion intensities were extracted using the default software settings: only peptides that were unique to a given protein or protein group were considered for quantitation. Fragment ion tolerance was set to 20 ppm for the most confident centroid peak, and the co-isolation threshold was set to 50%.

Data normalization was only performed during the TMT experiments. Therefore, the average of the total abundance of the 29 highest abundant proteins was calculated for all control samples. All samples were then proportionally scaled up or down based on their individual total abundance of the same 29 proteins. The proteins used for normalization were highly abundant in serum as well as in wound fluid samples, but none of them was a previously depleted highly abundant plasma protein.

The mass spectrometry proteomics data were deposited to the ProteomeXchange Consortium (http://proteomecentral.proteomexchange.org, accessed on 3 February 2023) via the PRIDE partner repository with the dataset identifier PXD039840, and these are publicly available as of the date of publication.

### 2.8. Quantification and Statistical Data Analysis

#### 2.8.1. Identification, Characterization, and Quantification of Common Wound Bed Proteome

The obtained data from the plasma and wound bed fluid specimens were log2 transformed and analyzed for a common set of proteins associated with both devices in the acute wound, as well as interaction with the plasma proteome by InteractiVenn [52]. Data were visualized using a principal component analysis with the ClustVis tool [53]. In the latter, unit variance scaling was applied to rows and SVD with the imputation used, in order to calculate the principal components. The proteins that were identified to be common to both devices were tested for enriched Gene Ontology (biological process, cellular compartment, molecular function), as well as Kyoto Encyclopedia of Genes and Genomes (KEGG) pathway terms using the Shiny GO software (version 0.76.3) [54]. Significance was tested with a two-sample test with a false discovery rate set to 0.05, according to Benjamini–Hochberg. The search tool for the Retrieval of Interacting Genes/Proteins (STRING v 11.5) database of physical and functional interactions was used to analyze the protein–protein interaction (PPI) of the selected proteins and clusters defined by kmeans (k = 4).

A statistical data analysis of the common wound proteome was carried out with GraphPad Prism (version 9.4.1). Mean values and standard deviations were calculated for each experimental condition or type of sample. The *p*-values between the samples were calculated by using an unpaired t-test per protein, with individual variances computed for each comparison, combined with the two-stage linear step-up procedure of Benjamini, Krieger, and Yekutieli. Significance was tested using a two-stage set-up method with a false discovery rate set to 0.01. For visualization by volcano plots, the l2 fc value was plotted against the −lg10 *p*-value by Manhattan distance in the *VolcaNoseR* web app [55]. Proteins were regarded as being differentially expressed when meeting the criteria l2 fc ≥ ± 1.5 and having an adjusted *p*-value of ≤ 0.01. Heatmaps were generated using the *ClustVis* [53] tool. The generation of tables was performed with Microsoft Excel 2018 (Microsoft Corporation). The generation of correlation plots was performed using GraphPad Prism (version 9.4.1).

#### 2.8.2. Identification and Characterization of Common Adsorbed Wound Bed Proteome on SMI Surface

The obtained abundances from the adhesive SMI proteome specimens were analyzed for a common set of proteins adsorbed to both devices by InteractiVenn [52]. Data were visualized through a principal component analysis with the ClustVis tool [53]. The adsorbed proteins that were identified to be common to both devices were submitted to Gene Ontology (biological process, cellular compartment, molecular function) as well as Kyoto Encyclopedia of Genes and Genomes (KEGG) pathway analysis by the Shiny GO software (version 0.76.3) [54]. Functional categories with an adjusted *p*-value of < 0.05 (Benjamini–Hochberg) were defined as significantly enriched. Heatmaps were generated using the ClustVis [53] tool. The generation of tables was performed with Microsoft Excel 2018 (Microsoft Corporation). The generation of correlation plots was performed using GraphPad Prism (version 9.4.1).

The statistical details of the experiments are presented in the relevant figure legends. A *p*-value of < 0.05 was considered significant. 

Significance: * *p* < 0.05/** *p* < 0.01/*** *p* < 0.001/**** *p* < 0.0001/ns = not significant.

## 3. Results

### 3.1. Patient Characteristics

We used a mass spectrometry approach based on Tandem Mass Tags (TMT) to quantify the common wound proteome formed around silicone breast tissue expanders (*SMIa*: SmoothSilk^®^ silicone mammary tissue expander (implant): surface roughness 4 µM; and *SMIb*: CPX4 silicone mammary tissue expander (implant): surface roughness 60 µM) from 1 to 5 days. *TMT* are isobaric chemical tags that provide multiplexing capabilities for the relative quantification of proteomics samples. 

As described in the Material and Methods, every patient received both expanders, with an applied randomized body site of implantation (Figure 1). The biological sample collection of wound bed fluid was performed from days 1 to 5 after expander implantation (wound collection drains; Figure 1b,c); the blood draws during hospital routine, namely during general anesthesia (intraoperatively); and expander surface stripping after 6 to 8 months (mean duration ± SD: 6.82 ± SD 0.86 months). During re-operation, the tissue expanders were exchanged with definite implants. Due to the intraindividual comparison of the tissue expanders, no matching of patient groups was carried out, and the comparative analysis was based on tissue expander surface roughness, as presented in Table 2. There were no differences in patient characteristics, mastectomy weight, or implant position (prepectoral, reconstruction volume, and intraoperative filling) between the two differently textured devices (Table 2) due to the implantation of both devices in every patient.

### 3.2. The Workflow of Proteomics Analysis

A label-free approach was applied to investigate the adhesion of wound proteome on the expander surface. A brief description of the experimental workflow of proteomics studies is shown in Figure 2, as explained in detail in the Materials and Methods. In brief, we acquired nanoLC-MS data on an FT mass spectrometer. The raw mass spectrometry files (MS) were analyzed and searched in the UniProt [56] human protein database using the Proteome Discoverer platform (Thermo Scientific). 

The proteins in the wound fluid and plasma samples were enzymatically digested, and the resulting peptides were labeled with TMT reagents (Figure 2a). To be able to compare the samples or subjects with one another, one-sixth of each digest was pooled and labeled separately. This pool served as the standard and was used for normalization. Subsequently, the wound bed fluid and plasma samples were combined into the TMT 6-plex sets of five samples from one patient/side (one subject; five samples (D1–D5); one tissue expander roughness type) and with one standard each (e.g., patient 1, WBF formed around 60 µM of CPX4 expander from days 1–5 post-surgery). The plasmas were assessed in a separate TMT 6-plex set. To increase the protein identification numbers, the pooled samples were chromatographically separated into eight fractions using a high-pH column before conducting the nanoLC-MS analysis. 

After removal of the devices in re-operation between 24 and 28 weeks after the initial expander implantation, six pieces of approx. 1 cm^2^ were removed from each SMI surface (*SMIa* and *SMIb*) and transferred into separate test tubes (Figure 2b). The proteins adhering to the silicone surface were then digested, with three pieces washed with PBS beforehand. From each of the six samples, 5% were then analyzed by nanoLC-MS. The quantification was based on the integrated peak areas per peptide, which are summed up at the protein level and result in a protein abundance. 

### 3.3. Quantification of Intraindividual Comparative Proteomic Profiling in Plasma, Wound, and SMI-Adhesive Proteome

The samples of wound bed fluid formed around the expanders and plasma were comparatively analyzed by the TMT-based proteomics method, in order to determine the proteins that may be related to the molecular mechanisms or triggers of an immediate inflammatory response towards silicone implantation in the first five days after surgery. We obtained a high coefficient of correlation (R2) value for the samples’ TMT log2 transformed abundance compared to the standard (Appendix A). Figure 3c presents the box plot for the log2 values of the TMT abundance ratios, corresponding to the wound bed fluid samples (*n* = 7; in triplicates) of each group (*n* = 3; plasma, *SMIa*, *SMIb*); this indicates that the interquartile range and median are similar between the biological replicates (*n* = 7) of each group (Figure 3c), showing the consistency of the values obtained in the LC-MS/MS measurements. 

We adopted a label-free quantitative proteomic analysis to investigate and identify which immediate foreign body response proteins from the wound environment potentially adhere to the SMI surface and chronically trigger excessive ECM turnover, resulting in fibrotic capsule formation. 

The principal component analysis revealed a high interindividual sample variation, by principal component 1 and principal component 2, which explained 43.2% and 21.1% of the total variance, respectively (Appendix A). The distribution of protein abundance displayed in Appendix A obeyed normal distribution and revealed a coefficient of correlation (R2) between the individual patient samples’ abundance ratios compared for both SMIs (Appendix A). Normality and lognormality tests confirmed a lognormal data distribution, due to failing normality tests with *p* **** > 0.0001 (Appendix A). 

### 3.4. Distribution of Wound Proteins Identified by Proteomic Analysis

A total of 942 plasma proteins as well as 1514 common wound proteins were identified in the wound environment around both expanders, all of which were matched in the UniProt_HomoSapiens database (Appendix A). Among the identified 1514 wound proteins, we characterized 895 as common plasma-derived wound proteome (Figure 3a, light blue; and Appendix A), confirmed by its abundance in the plasma serum sample. Six-hundred and nineteen wound proteins were not found in the plasma sample, although they were characterized as the local wound proteome (Figure 3a, light grey; and Appendix A).

The PCA analysis highlighted the spatial distribution of the 11 samples (plasma; WBF around 4 µM of textured expander 1–5D post-op; WBF around 60 µM of textured expander 1–5D post-op) and revealed segregation between the wound bed fluid and plasma samples, and high correlation between the two differential wound bed fluids formed around *SMIa* and *SMIb* at every sampling time point (Figure 3b). Specifically, the temporal visualization of wound proteome differentiation over the first five days after expander implantation revealed a periodical time-dependent proteomic shift towards a reduction in segregation between the plasma (Figure 3b; 0 h, blue) and wound proteome (Figure 3b; 24–120 h, *SMIa*: black, *SMIb:* grey); this shift was chronological, with wound proteome composition and abundance on day 5 reflecting a closer relation to pre-surgery plasma proteome.

### 3.5. Composition of Plasma-Derived Wound Proteome Formed around Silicon Tissue Expanders for the First Five Days after Implantation

To better understand the functions and pathways in which the proteins identified exclusively as plasma-derived wound proteome (895 proteins common to both devices) may be involved, we performed a (Figure 4a) Gene Ontology (GO) and (Figure 4b) Kyoto Encyclopedia of Genes and Genomes (KEGG) pathway analysis. The GO results revealed that the proteins of the plasma-derived wound proteome are mainly associated with the inflammatory and immune cell-activating BP (biological processes), such as neutrophil-mediated immunity (activation, regulation, degranulation), granulocyte and leucocyte activation, as well as the regulation of exocytosis and secretion by cells (Figure 4a). However, the most representative GO biological process category was secretion, encompassing 122 proteins; followed by leucocyte activation (103 proteins); and granulocyte, leucocyte, and neutrophil activation, as well as degranulation (Appendix A). 

The KEGG pathway enrichment analysis showed that the plasma-derived proteins in wound bed fluid that formed around the implanted tissue expanders (d1–d5) mainly belonged to complement and coagulation cascades, the proteasome, carbon metabolism, ECM–receptor interaction, metabolic pathways, phagosome, chemokine signaling pathway, spliceosome, focal adhesion, FC gamma R-mediated phagocytosis, and leucocyte trans-endothelial migration (Figure 4b and Appendix A). The complement and coagulation cascade was the most representative pathway, with an enrichment FDR of 2.6 × 10^−48^ encompassing 50 plasma-derived wound proteins, followed by immunity-related chemokine carbon metabolism and ECM–receptor interaction pathways, encompassing 29 and 20 expressed plasma-derived proteins, respectively.

Together, the GO and KEGG enriched functional annotation described an SMI-induced hyperinflammatory response with complement and immune cell activation, a cytokine storm, and coagulopathy. To further our understanding of the immediate change in wound proteome composition in the progress of capsular fibrosis, we evaluated the interaction of the 895 proteins, using the search tool to retrieve interacting genes/proteins STRING (V11.5; string-DB-org). The analysis provided us with a PPI network consisting of 868 nodes and 15,100 edges, with an average node degree of 34.8 and a local clustering coefficient of 0.1. The expected number was 5617 with a PPI enrichment *p*-value < 1.0 × 10^−16^ (Figure 4c). 

To further our understanding of the regulatory role of immediate wound proteome in the context of SMI-associated fibrosis, we applied K-means clustering and identified four major clusters of proteins (Figure 4c); these were mainly enriched in the KEGG pathways, reflecting different phases of tissue damage response (Table 3; cluster 3 and 4) and tissue repair (Table 3; cluster 1 and 2).

Tissue damage response was recognized in cluster 4 (blue), which included 308 proteins mainly enriched in complement and coagulation cascades, and in cluster 2 (yellow), with 159 proteins (cluster 4; Appendix A) involved in FC gamma R-mediated phagocytosis and leucocyte endothelial migration (cluster 2; Appendix A). Cluster 1 (red), containing 281 proteins with the most representative KEGG enrichment in the pentose phosphate pathway, glycolysis/gluconeogenesis, and carbon metabolism (cluster 1; Appendix A); and cluster 3 (green), which united 120 proteins involved in the core matrisome and/or ECM turn over (cluster 3; Appendix A), were recognized as being part of tissue damage repair and ECM turn-over processes.

Among the 895 common plasma-derived wound proteins, 437 proteins were identified according to their biological role (investigated in the UniProt database) as proteins involved in inflammatory excessive ECM turnover—the inflammatory matrisome. Depending on their annotated role (UniProt), we functionally annotated these into steps of tissue repair after SMI implantation (Figure 4d: step 1. clotting, step 2. inflammation, step 3. repair and fibrogenesis, and step 4. ECM turn-over). More details can be found in Appendix A.

A volcano plot was applied to delineate differential protein abundance against the corresponding *p*-value obtained (Appendix A). Among the 437 identified plasma-derived proteins in the wound bed, we identified a total of 102 that were significantly differentially upregulated over the total timespan, from day 1 to day 5 (Table 4 and Appendix A); these were common to both SMIs in comparison with the plasma of the patients and controls (Appendix A). Applied Manhattan distance clustering of the samples revealed the (i) discrimination between plasma and wound proteome, (ii) the proteome composition and expression similarity around both SMI devices, and (iii) discrimination between the day of sampling post-implantation.

Again, the wound proteins attracted to both SMIs from plasma showed the highest upregulation on day 1, with a gradual descent over the first 120 h post-implantation (Appendix A). 

Among the 102 differentially expressed proteins, we identified 42 proteins of the innate immune response (humoral and cellular), 34 proteins involved in the regulation of oxidative stress, 7 responders to mechanical stress, and 46 ECM proteins (Appendix A and Table 4). More details can be found in Appendix A. This set of 42 innate immune response proteins was composed of 29 factors of the humoral (DAPMs, PAMPs, AMPs, splicing factors, cytokines, and chemokines, etc.) and 13 proteins involved in immune cell activation and regulation. Among them, HSPA1B, A8, and A2 are implicated in the development of fibrosis and HSP90AA, a regulator of fibroblast activation (Figure 4e, Appendix A).

The set of ECM proteins was composed of 16 core matrisome proteins (13 ECM glycoproteins and 1 collagen chain) and 32 ECM-associated proteins (12 ECM regulators, 23 ECM-affiliated proteins, 6 cell-interaction components, and 3 secreted factors). Surprisingly, we found COL1A1, COL3A1 COL1A2, COL6A1, COL6A2, and COL6A3 to be abundant in the wound proteome around both SMIs; however, only COL1A, a prominent driver of fibrosis, was found significantly upregulated over the total 120 h post-surgery (Figure 4e, Appendix A). As expected, we identified the matrisome-associated S100A protein family as part of the FBR in the wound, with S100P, S100A4, S100A6, S100A7, S100A8, S100A9, S100A11, and S100A12 significantly upregulated compared with the corresponding unstimulated abundance in plasma. Conformingly, the upregulation of MMP98 and MMP9 enzymes, and EMC glycoproteins vimentin and nidogen in the wound milieu, reflect a clear change in ECM turnover (Figure 4e, Appendix A).

### 3.6. Proinflammatory Mediation in Local Wound Proteome Formed around Silicon Tissue Expanders for the First Five Days after Implantation

The plasma-derived wound proteome resembles a clear FBR towards both silicone implants in the first five days after implantation, characterized through different stages of tissue repair as an inflammatory matrisome.

In the next step, we aimed to further understand the functions of local wound proteome (Appendix A; 619 proteins common to both devices) and how the origin (plasma or local) of the FBR proteome can affect the progression from regular wound healing processes to tissue repair “gone wild” in the context of fibrosis. Again, we performed (Figure 5a) a GO and (Figure 5b) KEGG pathway analysis. 

The GO results revealed that the proteins of the local proteome were mainly enriched with immune cells activating BP (biological processes), corresponding to the plasma-derived wound proteome. However, the KEGG pathway analysis revealed a predominant enrichment in the proteasome category (Figure 5b) but not the complement system, as seen in the plasma-derived proteins (Figure 4b) in the wound bed fluid formed around the implanted tissue expanders (d1–d5). The STRING database was employed to construct the protein–protein interaction networks among these 619 proteins. The analysis provided us with a PPI network consisting of 601 nodes and 4341 edges, with an average node degree of 14.4 and a local clustering coefficient of 0.321. The expected number was 2635 with a PPI enrichment *p*-value of <1.0 × 10^−16^ (Figure 5c). As before, we applied kmeans (k = 4) clustering to the network into four main clusters, as described in Table 5.

Tissue damage response and repair were recognized and categorized, and found to be enriched in KEGG pathways comparably to plasma-derived proteome (Appendix A). Subsequently, 329 proteins were categorized as inflammatory matrisome proteins and functionally annotated into steps of tissue repair after SMI implantation (Figure 5d; step 1. clotting, step 2. inflammation, step 3. repair and fibrogenesis, and step 4. ECM turn-over). More details can be found in Appendix A. Among the 329 identified local wound proteins, we identified a total of 63 that were significantly differentially expressed (upregulated, Appendix A) over the total timespan, from day 1 to day 5 (Table 6), common to both SMIs of the patients and controls from day 1 to day 5 post-surgery (Appendix A). Corresponding to the plasma-derived wound proteome, the local wound protein samples also formed clusters (Manhattan distance) depending on the time-point of sampling post-implantation, and showed the highest upregulation on day 1, with a gradual descent over the next 4 days (Appendix A).

### 3.7. Wound Proteome Adsorption on SMI Surfaces in the First 8 Months Post-Implantation

SMI-associated capsular fibrosis is characterized by an excessive accumulation of the extracellular matrix as a response to different types of tissue injuries over a longer period. This FBR can be initiated by multiple and different stimuli and pathogenic factors, but mainly by the foreign body inserted, which triggers the cascade of reparation converging in molecular signals responsible for initiating and driving fibrosis. To answer the question of what part and which components of the implant encompassing immediate wound proteome serve as triggers of long-lasting chronic inflammation, we stripped the SMI-associated proteins from the tissue expander surface approx. 8 months post-op, as the expanders were exchanged with definite implants.

A total of 216 proteins adhered to *SMIa* and 230 to the *SMIb* surface (Appendix A and Figure 6a) were identified in the UniProt database (uniprot_HomoSapiens). The PCA variance analysis showed no proximity of the samples, revealing high interindividual variation in total adhesive proteomes (Appendix A). Among the adhered identified proteins, we identified 216 common proteins that were associated with both expanders (Appendix A and Figure 6a).

To gain a comprehensive understanding of the biological significance of these 216 silicone surface-associated and/or adhered proteins, we conducted a GO and KEGG analysis to characterize their potential functions. Our data revealed that these proteins are mainly associated with BP, such as immune cell secretion; the regulation of exocytosis; cell activation; and protein metabolism, targeting, and localization (Figure 6b). Moreover, concerning MF, these proteins are mainly involved in the binding of molecules to the surface: rRNA and RNA-, protein (cadherin, unfolded proteins, integrin, actin, cell adhesion, protein-containing complex, cytoskeletal protein, and signal receptor molecule) and lipid binding, as well as structural molecule activity in the context of ECM matrix structural constituent (Figure 6c). Moreover, in connection with the CC, the proteins were enriched for the collagen-containing extracellular matrix, extracellular matrix, and as an external encapsulating structure, as well as secretory granule and vesicle (Figure 6d). In addition, the results obtained from the KEGG pathway enrichment analysis confirmed that these silicone surface-associated proteins are related to pathways such as ECM–receptor interaction, complement, and coagulation cascades, and the phagosome, 8 months after device implantation (Figure 6e). Due to the high interindividual variation of protein abundance on the silicone surfaces, we compared mean protein abundance on *SMIa* and *SMIb* and revealed a differential pattern of adhesion of the common adhesive proteome on the devices (Figure 6f).

### 3.8. From Wound to Early-Stage Fibrosis: Adhesion of Inflammatory Matrisome to Silicone Surfaces

The final stage in wound healing/foreign body response to biomaterials is generally fibrosis and the fibrous encapsulation of the implanted device. Finally, we aimed to investigate and identify which proteins from the acute wound environment potentially adhere to the SMI surface long-term and chronically trigger excessive ECM turn-over 8 months post-silicon device implantation.

Corresponding to the wound proteome analysis, the 216 proteins of common (to both SMIs) SMI-surface associated proteome were categorized as inflammatory matrisome proteins and functionally annotated into steps of tissue repair after SMI implantation (Figure 7a,b and Appendix A). Among the common SMI-associated proteins, we identified three categories: adhered proteins from plasma-derived (102 proteins; Appendix A) and local (12 proteins; Appendix A) wound proteome, as well as novel SMI surface and time-point (8 months post-op) exclusive proteins (54 proteins, Appendix A). As presented in Table 7, we found that 21% of plasma-derived and 5% of local acute wound-associated inflammatory components adhered to SMI surfaces, from plasma mainly the complement system and local, the proinflammatory mediation by ribosomal proteins. 

Among the adhered core matrisome components, we found that not only do vimentin, vitronectin, fibronectin, laminin, fibrillin, elastin, glycoproteins, and proteoglycans participate in the formation of extracellular matrix (ECM), but they are also associated with the SMI surface. Additionally, we identified various plasma derived collagen types (Appendix A; COL-1, -3, -4, -5, -6, -12, -14, -15, -18), although only COL1 and COL6 were significantly upregulated in the acute wound (Figure 7c(ii)). In the context of chronic progression from an acute wound to capsular fibrosis, our data reveal that molecular proinflammatory mediators, such as (i) TGFß1, CD44, HSP70, and HSP90 family members, as well as fibrosis drivers from (ii) core matrisome components ECM1, fibronectin, vitronectin and vimentin, COL1A (A1–A2), and COL6 (A1–A3); but also (iii) secreted factors of the S100A (A4, A10, and A11) family, are recruited as plasma components to the wound and associated/adhered to the SMI surface (Figure 7c). COL3 was significantly upregulated only on days 3 and 4 and dropped to plasma levels on day 5 (Appendix A). At these time points, COL3 was significantly increased in comparison to COL1 (Appendix A). Generally, the expression of these proteins periodically decreased during the first five days post-op, except TGFß1 (Figure 7c(i)).

Additionally, local wound exclusive profibrotic response, HSP60, as well as MCP-1, were found among the adhered proteins (Figure 7d). Both act directly on antigen-presenting cells (APC) and enhance T-cell activation. Importantly, we found T-complex (in the form of CCT2) abundance/adhesion on the SMI surface (Appendix A). No detection of Mammaglobin A on the SMI surface could be confirmed.

Among the 54 proteins exclusively found 8 months post-implantation on the SMI surface, we identified a total of 37 proteins that were involved in the innate response; these were mainly proinflammatory- and profibrotic-mediating 40S/60S ribosomal proteins; the MIC complex and macrophage-capping protein. The most interesting among the matrisome components was the detected CD9 antigen—the pro-fibrogenic and proinflammatory macrophage senescence factor, confirming the phagocytotic activity seen previously in GO and KEGG enrichments (Figure 6b,c).

## 4. Discussion

### 4.1. Intraindividual Comparative Proteomic Profiling in Plasma, Wound, and SMI-Adhesive Proteome

Any implanted foreign body elicits an immediate inflammatory response resulting in either fibrous encapsulation by excessive ECM turn-over or incorporation of the implanted biomaterial [57]. The controlled method of sample collection, as well as the analysis used, is an integral step in biological significance in the diagnostic research process of capsular fibrosis etiology. 

Blood biomarkers are widely applied for the screening and diagnosis of inflammation because of their availability. However, the extent of the soft tissue reaction surrounding the implant depends on several factors, although to study wound infection directly, wound bed fluid is the prerequisite sample to choose. Wound bed fluid is not only tied to the wound bed environment but is easily available. Proteins in wound fluid obtained by a sterile, closed-suction drain placed in the subcutaneous tissue (following mastectomy) [47] are not only exposed to proteinase 3 [58], but also to plasminogen activity [59]. Moreover, drain collection from day 1 to day 5 post-implantation in flasks stored at room temperature establishes a 24 h time window of proteomic degradation. 

The identification of protein repertoires in the wound bed fluid directly after SMI implantation is notoriously challenging for the discovery of potential biomarkers for the switch and progression from the normal healing response towards chronic fibrosis and the excessive encapsulation of the silicon device. Moreover, protein adsorption at the implant surface is a key driver of local reactions to silicone; thus, we aimed to identify proteins that surround the device non-specifically in the acute wound and adhere to the silicon surface.

In this study, we investigated the proteomes of pre-operative plasma (−1 d), wounds that are derived from surgical drainages following NSME (d1–d5) and associated with/adhered to the SMI surface approx. 8 months post-op. 

The etiology of this exaggerated capsule formation is multifactorial but primarily induced by immune mechanisms towards the foreign material silicone. Performed in vivo for the first time in human patients, we decided to intra-individually identify, characterize, and quantify the global FBR against SMI, common to two commercially available tissue expanders with varied surface roughness and topography: the *SMIa* with reduced surface roughness (4 µM of Ra; SmoothSilk^®^, Motiva) and conventional *SMIb* (60 µM of Ra; CPX^®^ 4, Mentor). Sample collection was performed simultaneously for both devices in a sterile environment from the same individuum at the same time point (*n* = 7) (Figure 1 and Figure 2). Here, we provide insight into a set of diagnostic biomarkers applicable to silicone devices with varying surface topography roughness.

### 4.2. Immunomics: The Essence Lies in Sample Integrity

Investigation of the wound proteome revealed an SMI common, topography-independent, as well as topography-exclusive wound and adhesive proteome (Figure 3a and Figure 6a). We show a high-abundance correlation between the biological and technical replicates of the wound fluid and tissue expander surface samples (WBF: Figure 3c, Appendix A; and SMI surface-adhesive proteome: Figure 6b and Appendix A), validating our intra-individually analyzed dataset. Additionally, there were few differences regarding the total spectral count between groups, validating the sample preparation and allowing for semi-quantitative comparison between plasma and plasma-derived wound proteome.

Corresponding to previous studies [48], we found significantly higher protein numbers in wound bed fluid compared to plasma (Figure 3a). Our analysis demonstrated a high interindividual sample variance and intraindividual composition between plasma and acute wound fluids, as well as a collection time-point variation within wound samples from NSME drainages collected 1–5 days post-surgery (Figure 3b). Furthermore, the protein substrate environment varied between groups, as the wound fluid samples contained a more varied environment compared to plasma (Figure 3b). 

As neutrophil proteases are acquired in a dose-dependent manner from activated neutrophils during inflammation [60], we hypothesized that the interindividual sample distribution and variation could be influenced by differences in the respective protease environments due to the collection time-point. The interindividual differences seen in the SMI-common wound and adhesive proteomes are most likely due to individual changes in substrate abundance and protease expression [48,61]. 

Taken together, these results serve as proof of the principle that the intraindividual comparison and analysis of proteomes can indeed withstand detectable interindividual differences in common protein expression and is crucial for the immunoreactivity data comparison of SMIs with varied topographies, in a general overview of silicone device immunoreactivity.

### 4.3. Immediate Inflammatory Rush in the Wound after SMI Implantation

The characterization and quantification of inflammatory composition and kinetics in wound bed fluid and the verification of adhered wound proteome on the SMI surface proved a valuable strategy for the possible development of diagnostic tests for subclinical capsular fibrosis.

The common wound bed proteome was composed of a plasma-derived and local wound protein fraction generated around both devices (Figure 3a). The quantification of plasma-derived proteins in wound fluid showed significantly higher protein expression in wound bed fluid post-op compared to plasma pre-op, which progressively decreased to pre-op levels over the first five days after device implantation (Figure 3b,f and Appendix A). We also analyzed the wound proteome derived from local tissues and demonstrated the same progressive reduction in protein abundance, respectively (Figure 4e, Appendix A).

For the first time, in a clinically synchronized controlled setting, intra-individually in humans, with fresh, sterile, time-point-specific collected samples, our results significantly expand the number and origin classification of inflammatory wound bed proteome. In the present investigation, DEPs with high significance in wound fluid and adhered to SMI—such as the HSP90 family, collagens, and S100A family—are actively associated with phases of disturbed wound healing (clotting, inflammation, fibrotic repair, and excessive ECM turnover) and can be further characterized as inflammatory matrisome.

### 4.4. Pathogen Binding and Activation of Inflammasome in the Wound

Immediately following the placement of a breast implant, the destruction of the tissue initiates the host response to reactive oxygen species (ROS), nitric oxide, and toxic and mechanical stress [11,29]. The first remarkable result to emerge from the data is 34 plasma-derived DEPs with high significance, such as superoxide dismutase (periredoxin, extracellular and mitochondrial) and catalase (Appendix A), which are actively involved in the oxidative response to NO and ROS. However, we found no expression of additional ROS or NO disbalance responders in the local wound compartment.

The increased expression of plasma-derived PAL-1 during the first five days after implantation was observed. This factor plays an essential role in blood coagulation, by inhibiting the conversion of plasminogen to plasmin, thereby preventing fibrinolysis [62,63]. Moreover, we observed elevated levels of plasma-derived HBB, HBA2, MB, HBD, HBG1, and, locally expressed in the wound, HBG2, which reflects the first response to tissue damage, bleeding, and clotting. Damage to the epithelial cells leads to the aggregation of erythrocytes and platelets that form a blood clot to contain the spread of damage.

The destruction of the tissue and accumulation of altered self-components as well as reactions to noninfectious foreign material triggers inflammation and the release of damage-associated molecular patterns (DAMPs), or allows the entrance of microorganisms and thereby pathogen-associated molecular patterns (PAMPs) [29]. To our surprise, we found that PGLYRP1 was differentially expressed in the local environment. This 196aa innate immunity protein acts as a bactericidal pattern receptor that binds to murein peptidoglycans (PGN) of Gram-positive bacteria [64]. Moreover, in the present investigation, we could confirm the abundance of LBP, BPI, CFL1, CAMP, LYZ, and B2M in the wound. Fifteen plasma-derived DEPs with high significance, such as CAMP, LTF, DEFA1B, and ITGB2, are actively associated with host–pathogen interaction and antimicrobicidal response. CAMP, a 177aa cationic antimicrobial protein with a molecular mass of 18kDa, is an acute-phase reactant that binds to bacterial lipopolysaccharides (LPS) [65,66] and acts via neutrophil N-formyl peptide receptors to enhance the release of CXCL2 [67].

### 4.5. FBR and Inflammation: The Response in Inflammatory Matrisome

Local cells such as dendritic cells and macrophages are activated by DAMPs and PAMPs through toll-like receptors (TLRs) and protease-activated receptors (PARs), and they produce the first round of pro-inflammatory mediators [11]. Interestingly, thePGRP1, which is detected in the local wound environment, acts in complex with the Ca^2+^-binding protein S100A4 as a chemoattractant that is able to induce lymphocyte movement [68]. The abundance of plasma-derived S100A4, as well as differentially expressed CD44 and MNDA, could be confirmed in the wound environment (Appendix A).

In our study, strong evidence was found in the GO enrichment analysis of plasma-derived and local wound proteins, revealing highly significant involvement in the biological processes of neutrophil, granulocyte, and leucocyte activation, mediation, and degranulation (Figure 4a), which confirms an infiltration of neutrophils and monocytes that respond to the pro-inflammatory stimuli at the implant surface. Phagocytes such as neutrophils and macrophages remove tissue debris and potentially threatening particles, whereas neutrophil and granulocyte degranulation induce the first inflammatory storm [11,29].

The granules of neutrophils contain numerous enzymes, such as matrix metalloproteinases (MMPs), elastase, and cathepsins that can specifically cleave collagenous and non-collagenous connective tissue components that are mandatory for pro-fibrotic ECM turn-over. Correspondingly, we identified plasma-derived DEPs involved in ECM remodeling, such as ELANE, leucocyte elastase inhibitor (SERPINB1), and NCF1B, as well as MMP2, MMP8, and MMP9 (Appendix A). Moreover, neutrophils play an indirect role by activating further innate cellular components [30,60], reflected in highly significant DEPs such as CSF1R and BCAP31 (Appendix A).

In a pro-inflammatory environment, activated monocytes and resident macrophages(M1) produce pro-inflammatory mediators such as IL-6, TNF-α, and IL-1 [69]. We detected IL1RN, IL1RAP, IL1R2 in plasma (Appendix A), and IL6ST in local wound proteome (Appendix A). Our findings confirm the presence of activated macrophages in the wound environment that produce IL-1 to activate fibroblasts and induce the overproduction of ECM proteins. Although we did not identify TNF-α components in the wound proteome, we detected TNF-α-triggered signal transduction via the NF-κB pathway, which was reflected in the local expression of tumor necrosis factor receptor superfamily member 10C and NFKB1 (Appendix A). 

Remarkably, traces of antifibrotic effected by interferon (IFN)-γ produced ls were visible in plasma-derived and local interferon-inducible components IFIT3 (Appendix A), IFIT2, IFI30, IFI35, and IFI16 (Appendix A). Effective healing is usually characterized by a dominant T helper 1 (Th1) cell response, whereas a predominant T helper 2 (Th2) response and an increase in T helper 17 (Th17) cells lead to chronic inflammation, which can ultimately result in fibrosis [29]. On the other hand, Th2 cells mediate the adaptive immune response to injury by producing pro-fibrotic (anti-inflammatory) cytokines (e.g., IL-4, IL-13, IL-10); Th1 cells mediate tissue damage response by Th1-related proinflammatory cytokines (IFN-y, IL-12) that suppress fibroblast-induced collagen synthesis and attenuate fibrosis [70]. In addition, IFN-γ production up-regulates the expression of matrix metalloproteinases (MMPs), whose proteolytic activity helps alter ECM remodeling. As a commonly recognized opponent of Th1 cells, Th2 cells can alter Th1-associated IFN-γ expression levels, and high levels of Th2 cytokines have been reported in several fibrotic diseases [70]. 

Taken together, we obtained comprehensive results demonstrating two main implant-induced pro-fibrotic progressions—a systemic (plasma-derived) proinflammatory response and local proinflammatory mediation in the wound. The wound bed proteome analysis post-silicon implantation provides detailed insights into wound healing stages characterized by various inflammatory matrisome proteins. 

### 4.6. From Wound to Fibrosis

In the course of wound healing, inflammatory responses subside and turn into repair responses through the effects of anti-inflammatory and pro-repair cytokines. In accordance, we found transforming growth factor-beta-induced protein (TGFB1, Appendix A) as a compound of the plasma-derived protein and locally expressed late-transforming growth factor binding proteins LTBP4 and LTBP2, as well as LTBP1 (Appendix A), the latter being differentially expressed over the first five days after SMI implantation (Figure 5g, Appendix A). LTBPs are involved in fibroblast proliferation and migration; thus, with the finding of plasma-derived SLC9A3R1 and local CRYAB, we provide further evidence for the stage of fibrogenesis and fibroblast proliferation into the more contractile myofibroblasts [29]. Myofibroblasts produce an extracellular matrix (ECM) to close the open wound and form a scar [71]. As already reported, active ECM turnover could primarily be suspected through the detection of MMP2, 8, and 9, whereas MMP8 was upregulated in the wound over the total first five days post-surgery. Further, we identified and characterized 71 plasma-derived and 30 local wound proteins as core matrisome, as well as 142 plasma-derived and 48 local wound proteins as matrisome-affiliated proteins [72,73]. Strikingly, we could identify not only pro-fibrotic marker COL1A (plasma-derived; Appendix A) and ELANE as being upregulated over the first five days post-op, but also the fibrosis drivers S100A8 and S100A9 derived from plasma. Moreover, COL6 and COL3 were detected; however, these were not significantly increased over the total 5-day period (Appendix A). The set of profibrotic markers was supplemented with the locally expressed keratins (Appendix A: KRT-8, -36, -82, -75, -76 and -79), as well as collagen alpha chains (Appendix A: COL-12A1, -14A1, -15A1, and -18A1). 

Regularly, in the last phase of repair excess, ECM is degraded for the incorporation of new cells as the tissue regains its structure and function. However, in fibrosis, aberrant wound healing results in the excessive accumulation of collagen and ECM components.

To identify potential biomarkers for the switch and progression from the normal healing response towards chronic fibrosis and the excessive encapsulation of the silicon device, we further focused not on serum, as previously reported [20,74], but on wound bed fluid proteins associated with both silicone surfaces and that were still adhered to them after 8 months lingering in the body under pro-inflammatory and pro-fibrotic conditions.

Our data reveals not only a common SMI-associated proteome (Figure 6), independent of surface topography, after 8 months residing in a human body, but also a common implant-enclosing wound proteome adhered to SMI surfaces (Figure 7). Enrichment analyses of the common (on *SMIa* and *SMIb*) adhesive proteome confirmed the main involvement in ECM turnover reflecting early-stage fibrosis.

Aside from the wound-derived proteome, we identified an additional 54 proteins, which were found to be adhered to SMI surfaces but not detected in wound bed fluid. This early-stage fibrotic set of proteins was characterized by several 60 S and 40 S ribosomal proteins, a major group of proinflammatory mediators [75,76,77]. Moreover, we detected CAPG and CD9 on SMI surfaces. This favorably correlates with a recent study in fibrotic adipose tissue, where Rabhi and colleagues [78] showed that CD9^+^ senescent macrophages activate a fibrotic transcriptional program in adipocyte progenitors.

Strikingly, among the SMI adhesive proteome, we found various collagens and S100A family members (Appendix A), all previously identified as part of the plasma-derived inflammatory matrisome in the acute wound. In healthy epidermal tissue, collagen type I is more prevalent compared to type III, which comprises about 30% of the total collagen content [79,80]. Due to ECM turnover during wound healing, COL3 can increase up to 90% and a decrease in type III can lead to excessive scar formation [71]. In contrast with earlier reports, we did not observe a significant upregulation of COL3 but of COL1 and COL6 in the acute wound (Appendix A). However, COL3 was detected on the SMI surface, significantly upregulated on days 3 and 4 post-op, and we found higher levels of COL3 compared to COL1 in the acute wound proteome (Appendix A). Here we manifest the role of COL1 in late fibrosis [81,82], moreover, our new data may also provide information on the chronological production of other collagens and non-collagenous proteins.

Unexpectedly, S100A8/A9, a major fibrogenesis marker and fibrosis driver (fibroblast proliferation, differentiation, and activation of collagen production) [83] was found only in the wound environment, whereas S100A4, A10, and A11 were found to be associated with the silicon surface (Figure 7 and Appendix A). The latter was not surprising, as S100A4 was identified as a fibrosis driver and a useful biomarker for diagnosis and monitoring disease progression [68]. Our experiments do not only corroborate with previous studies; here, we provide the finding of different roles for the S100A family members in fibrosis, as for example S100A8/9 marks the acute inflammatory profibrotic process, and S100A4 exerts its pro-inflammatory role in early-stage fibrosis 8 months after SMI implantation. Furthermore, multiple processes involved in fibroblast activation on the SMI surface were reflected by the confirmation of HSP90 adhesion to the implant site (Figure 7 and Appendix A). HSPs are involved in the activation or inhibition of epithelial-mesenchymal transition signaling pathways and the generation and degradation of the extracellular matrix to regulate fibrotic diseases; thus, we were not surprised to find HSPA1B, HSPA1A, HSPA8, HSPB1 (plasma-derived; Appendix A), and HSPD1(local; Appendix A) attached to the surface [21]. 

Moreover, we can demonstrate a chronic fibrotic signaling axis, with PGLYRP1, part of the ant microbicidal response in a wound, that forms an equimolar complex with heat shock protein HSPA1A (on expanders) and activates the TNFR1 receptor (expressed locally TNF superfamily 10) from the immediate inflammatory response to chronic inflammation 6–8 months after SMI implantation.

Strikingly, HSP60, a part of the antimicrobial inflammatory response [84] in the acute wound, was found attached to the silicon shell surface. The stress-induced protein is involved in the bystander activation of T cells. By inducing secretion of the proinflammatory cytokine IFN-γ [85], it acts directly on activating bacterial HSP-responsive gamma delta T cells [84]. It can also act as a target for autoreactive hsp60-specific T-cell responses and directly contribute to chronicity at the site of inflammation. Hsp60-responsive T cells are pro-inflammatory cells with an increased secretion of IFN-γ [85] and low IL-10 profile. Involved in the human chronic inflammatory diseases of Th1 and Th2 cells, it can majorly distract wound healing [86]. Our values correlate favorably well with a series of reports and further support the idea that HSP60 is an essential homeostatic antigen with both immunoregulatory and inflammatory properties. Its abundance confirms the presence of antimicrobial antigens and response in the wound after SMI implantation. However, the signaling of an endogenous protein expressed in low amounts chronically attached to SMI surfaces may lead to distinct patterns of activation of TRL-expressing cells, such as dendritic cells, macrophages, and T cells.

Derived from wound proteome, the detection of CCT8 (local) and T_REG_ receptor CD44 (plasma-derived), both adhered to SMI surfaces (Appendix A), clearly confirm T cell response at and directed to the implant site. Only scarce data about specific local side effects (local immune response, activity of immune cells) focusing on lymphocytes isolated from fibrous capsules have been reported so far. In a previous attempt, we characterized the cellular composition of fibrous capsules formed around SMIs, by showing that macrophages and fibroblasts were the most predominant cell populations in the region abutting the silicone surface (designated as “pseudo synovium”) [21]. Strikingly, among T cells, Treg numbers, in peri-SMI fibrotic capsules, were inversely proportional to the degree of fibrosis (Baker scores I to IV). Most interestingly, we showed that Tregs were decreased in those capsules removed from patients with clinically severe symptoms of capsular contracture (Baker scores III to IV) [21].

Thus, we deciphered the three-dimensional composition of a silicon implant surface associated/adhesive proteome (Figure 8) by identifying the adhesion of (i) plasma and local tissue-derived proteins in the acute wound and (ii) components expressed later in the early stages of fibrosis. Among these, we identified long-term capsular fibrosis markers after a simultaneous NSME and implant-based breast reconstruction, providing novel diagnostic targets for the long-term tracking of capsular fibrosis and potential capsular contracture.

As also shown here, the wound and SMI-adhesive proteomes of individuals vary greatly; therefore, investigating specific differences in the groups would benefit from larger sets of data to exclude unreliable findings and intensify the differences. Nevertheless, the fact that significant differences still emerged between the different proteomes, despite the limited patient number, illustrates the power of the intraindividual comparison of two different silicone breast implants in detecting subtle qualitative differences in protein patterns. Moreover, the detection of mammary gland-specific breast cancer marker Mammaglobin A in local wound tissue after bilateral mastectomy in all tested breast cancer patients (no tumor diagnosis yet) strengthens our data integrity immensely and confirms the high specificity but applicability of our approach. Although limited, we demonstrate the power of our combined proteomics and bioinformatics approaches in a detailed immunomic picture of capsular fibrosis etiology, from the immediate inflammatory response to early-stage fibrosis.

Finally, we would like to highlight our “secondary finding” of the SMI surface topography exclusive acute wound and adhesive proteome. Biomaterial surface chemistry, mechanical properties, and topography have been shown to influence the immune response [17,33] and implant surface-associated biofilm formation, especially due to antimicrobic (antibiotic)-resistant microbial strains, which can lead to chronic immune system activation [87,88]. However, depending on the topography of these surfaces, varying degrees of capsular contracture have been reported [22,89,90], and recent studies in rodents and preserved capsular tissue samples have confirmed a reduction in inflammation and foreign body response on implants with an average roughness of 4 μm [50]. These results propose a further investigation of the effect of surface topography on microbiome and proteome composition after breast implant insertion in a controlled clinical setting with breast cancer patients, as demonstrated here. 

Our innovative, intraindividual, comparative molecular identification of biomarkers and their chronological progression generates information of great importance for a basic understanding of the fibrotic side effects of SMIs, in terms of diagnostics prevention, potential new therapeutic approaches, as well as the improved biocompatibility of SMIs.

## 5. Conclusions

Our study comprehensively portrayed serum, wound fluid, and SMI surface adhesion proteomic profiling in patients after simultaneous prophylactic NSME and breast-tissue expander (implant)-based reconstruction. We provide insights into the composition of the wound proteome, comprised of a systemic plasma-derived and a local (expressed in local tissue) wound proteome, the first serving as an inflammatory foreign body responder, and the second as a proinflammatory mediator. We found massive antimicrobicidal activity and a complex inflammasome immediately after SMI implantation. This indicates microbial colonization and biofilm formation as potential FBR triggers and needs to be investigated in more detail.

By analyzing fresh human samples collected in a controlled clinical setting, these results offer unique evidence for an immediate systemic FBR burst response with a subsequent decline during the first days, as well as proinflammatory mediation from the local tissue in the wound during the same period and in the long-term, as adhesive and resident inflammatory matrisome on the SMI surfaces. Our data revealed potential early markers, such as S100A8/A9, and potential long-term markers, including COL1, HSP90, S100A4, and ELANE, with high diagnostic sensitivity. Furthermore, the presented results provide preliminary information on DEPs in wound fluid and their possible application for discovering new therapeutic targets to halt the development of capsular contracture, as well as for the identification of further candidate biomarkers of this disease. Therefore, this study and our analytical approach warrant further investigations to comprehensively and conclusively elucidate the role of these proteins in the pathogenesis of capsular fibrosis in implant-based breast reconstruction. 

## Figures and Tables

**Figure 1 biomolecules-13-00305-f001:**
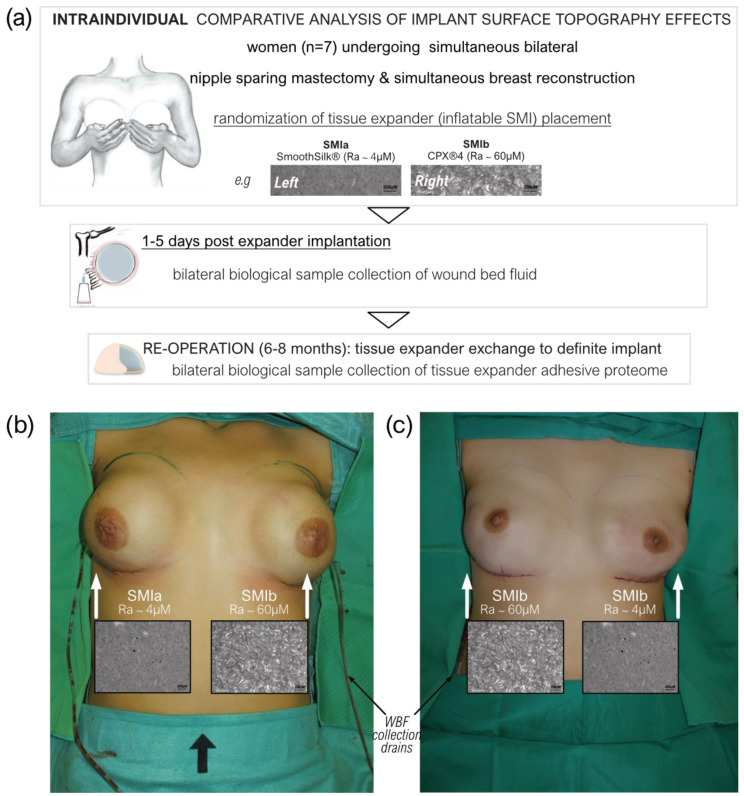
Standardized intra-operative photo documentation of a bilateral tissue expander-based breast reconstruction. (**a**) Each patient received both types of expanders, the novel surface-roughness reduced SmoothSilk^®^ breast expanders (Motiva Flora^®^, Establishment Labs, Costa Rica: surface roughness ~4 µM of Ra; termed *SMIa*) and the routinely used CPX^®^4 breast expanders (MENTOR^®^, USA: surface roughness ~60 µM of Ra; termed *SMIb*), randomized to the left or right breast after bilateral prophylactic NSME. (**b**) Patient 003; **Right**: *SMIa* (SmoothSilk^®^), **Left**: *SMIb* (CPX^®^4). (**c**) Patient 001; **Right**: *SMIb* (CPX^®^4), **Left**: *SMIa* (SmoothSilk^®^).

**Figure 2 biomolecules-13-00305-f002:**
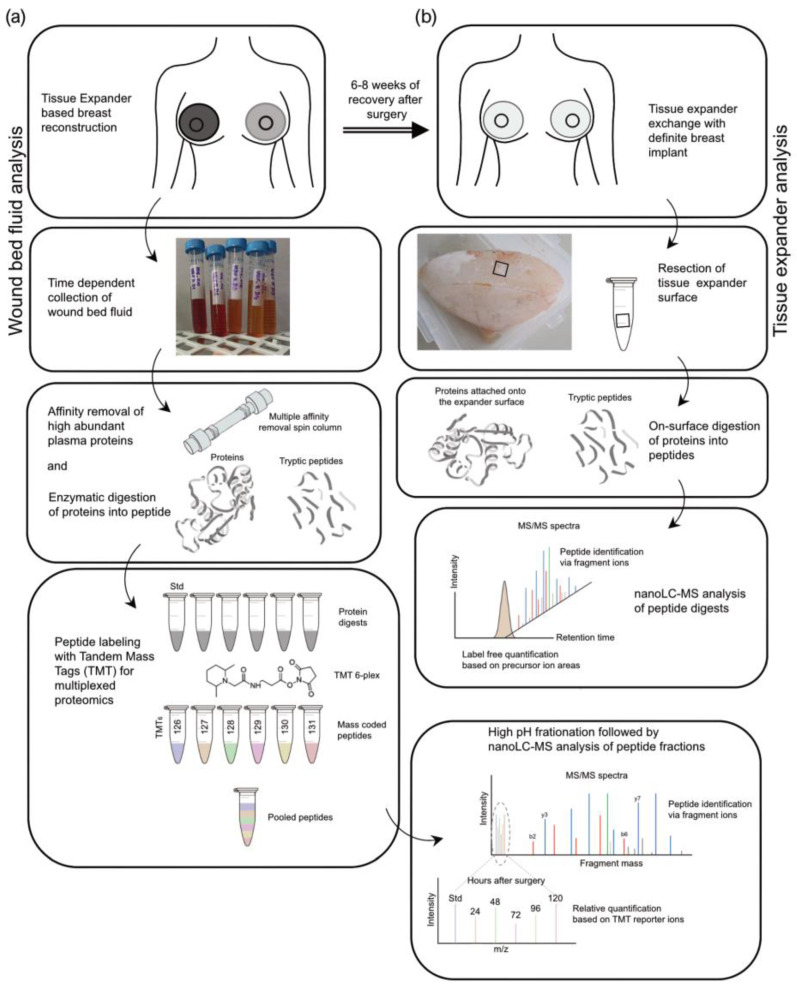
Experimental design of proteomic analyses. Stages (black hollow square) of sample preparation and proteomic analysis. (**a**) Wound bed fluid analysis. (**b**) Tissue expander analysis.

**Figure 3 biomolecules-13-00305-f003:**
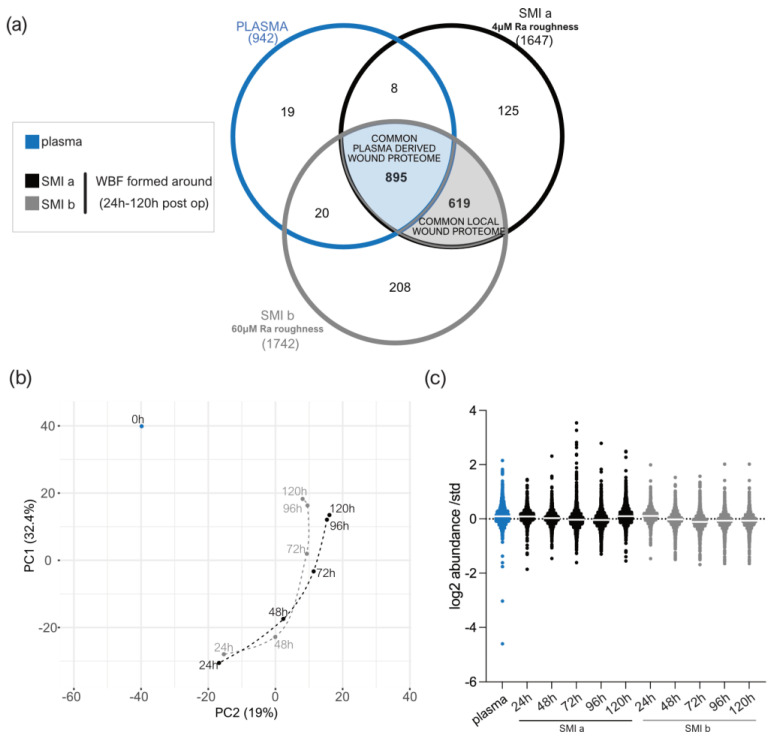
LC−MS/MS reproducibility and distribution of proteins identified. (**a**) Venn diagram showing the distribution of proteins in the collected plasma and wound bed fluid formed intra-individually around tissue expanders with 4 µM (SmoothSilk^®^) and 60 µM (CPX^®^4) surface roughness. (**b**) PCA was obtained with a median log2 abundance of the proteins found in all samples. (**c**) Scatter plot of median results with interquartile range obtained by wound proteome analysis on 7 patients, with 2 different silicone tissue expanders implanted, in plasma and wound bed fluid formed around the devices 1 to 5 days post-surgery.

**Figure 4 biomolecules-13-00305-f004:**
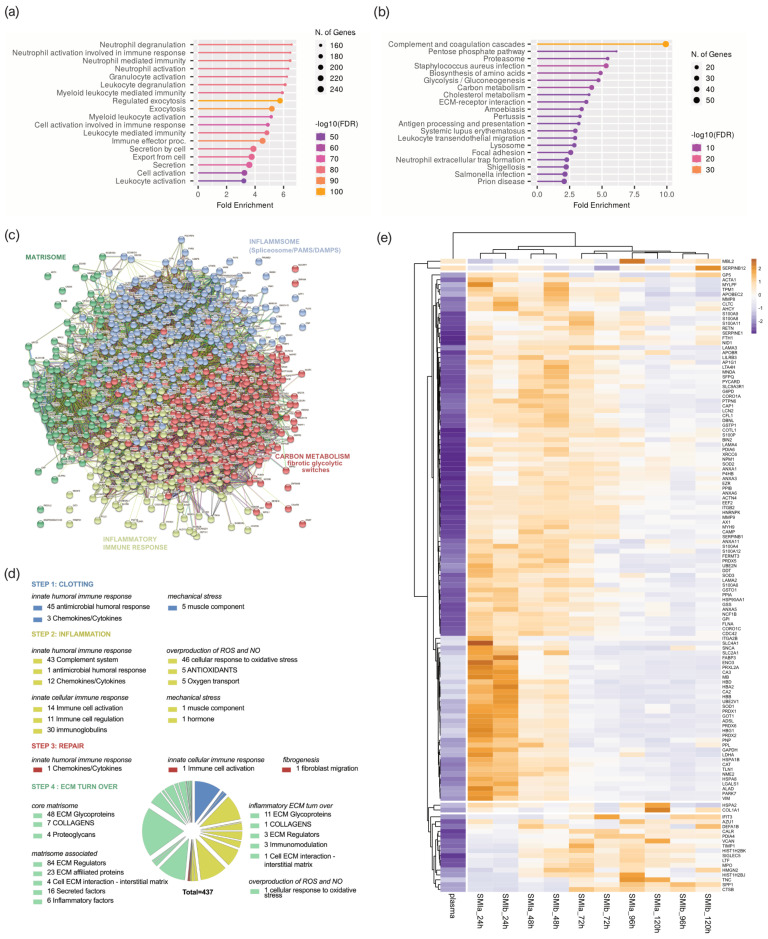
Plasma-derived wound proteome. (**a**) GO biological process enrichment, (**b**) KEGG pathway enrichment analysis, (**c**) protein–protein interaction regulatory network by K−means clustering based on STRING database, (**d**) functional categorization in the context of tissue repair of common plasma-derived wound proteome formed around 2 different silicone tissue expanders implanted (intraindividual comparison; *n* = 7), 1 to 5 days post-surgery. (**e**) Heatmap analysis of DEP in the plasma-derived wound proteome. *Rows:* Clustered by Manhattan distance, average method, and tightest cluster first tree ordering. Columns: Clustered by correlation distance, average method, and tightest cluster first tree ordering.

**Figure 5 biomolecules-13-00305-f005:**
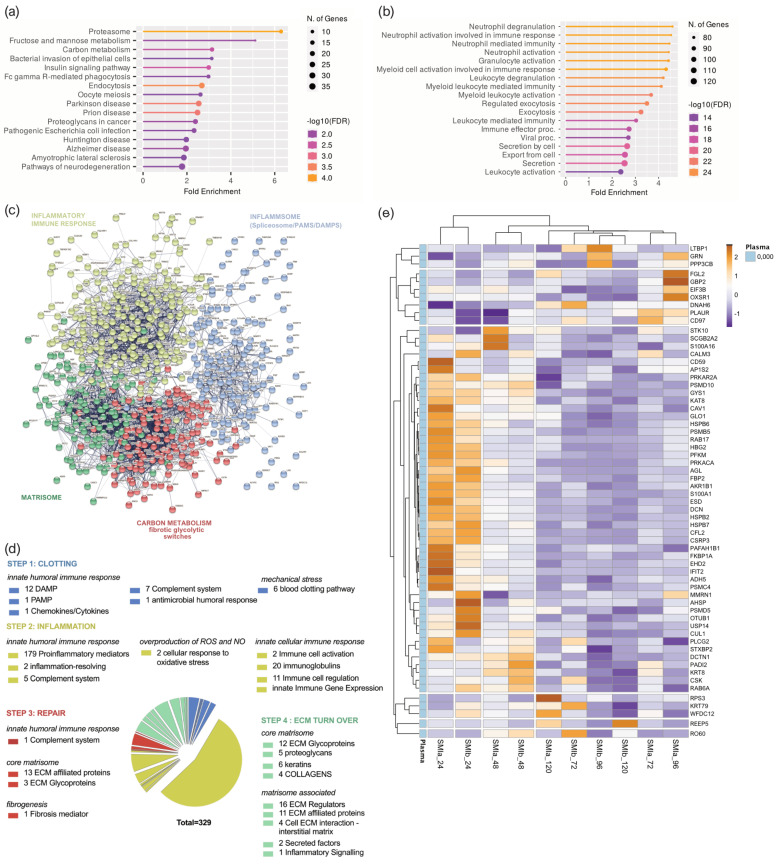
Local wound proteome. (**a**) GO biological process enrichment, (**b**) KEGG pathway enrichment analysis, (**c**) protein–protein interaction regulatory network by kmeans clustering based on STRING database, (**d**) functional categorization in the context of tissue repair of common local wound proteome formed around 2 different silicone tissue expanders implanted (intraindividual comparison; *n* = 7), 1 to 5 days post-surgery. (**e**) Heatmap analysis of DEP in the local wound proteome. *Rows:* Clustered by Manhattan distance, average method, and tightest cluster first tree ordering. Columns: Clustered by correlation distance, average method, and tightest cluster first tree ordering.

**Figure 6 biomolecules-13-00305-f006:**
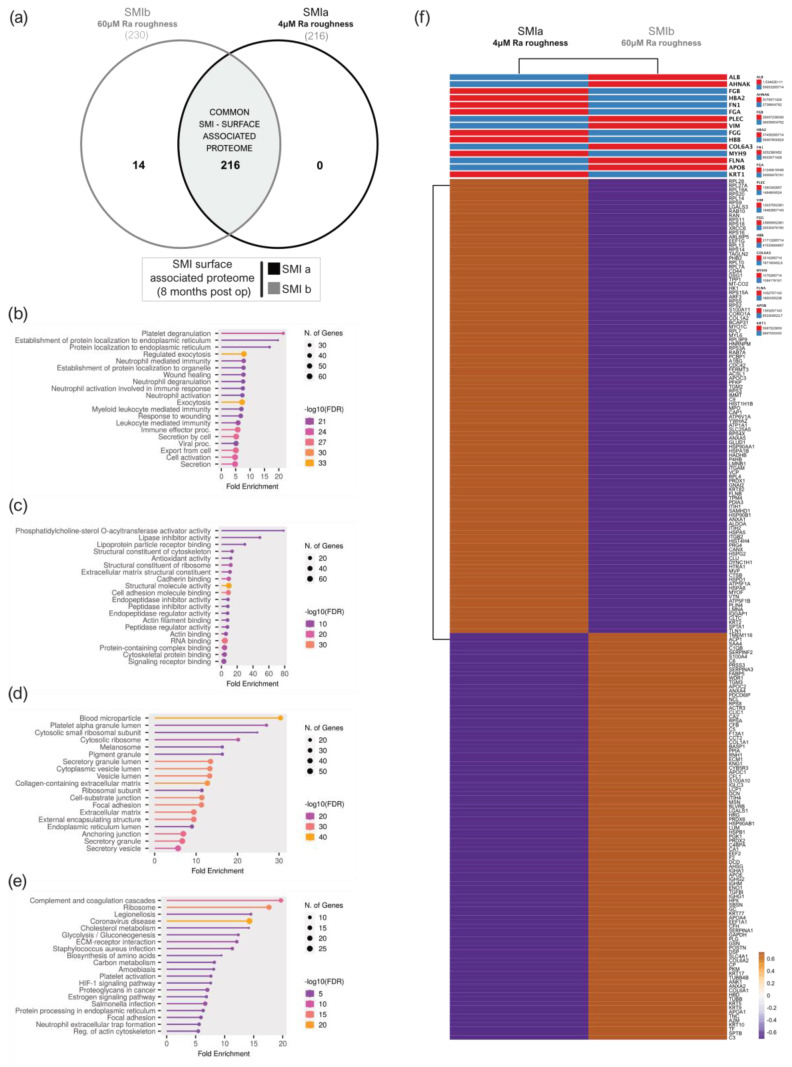
SMI surface-associated adhesive proteome. (**a**) Venn diagram showing the distribution of proteins associated at silicone tissue expander surface (*SMIa*: 4 µM and *SMIb* 60 µM surface roughness. (**b**) GO biological process enrichment, (**c**) GO molecular function enrichment, (**d**) GO cellular component enrichment, (**e**) KEGG pathway enrichment analysis of common SMI surface-associated adhesive proteome. (**f**) Heatmap analysis of mean protein log2 transformed abundance of common adhesive proteome on *SMIa* and *SMIb*. Rows are centered; unit variance scaling is applied to rows. Both rows and columns clustered using correlation distance and average linkage.

**Figure 7 biomolecules-13-00305-f007:**
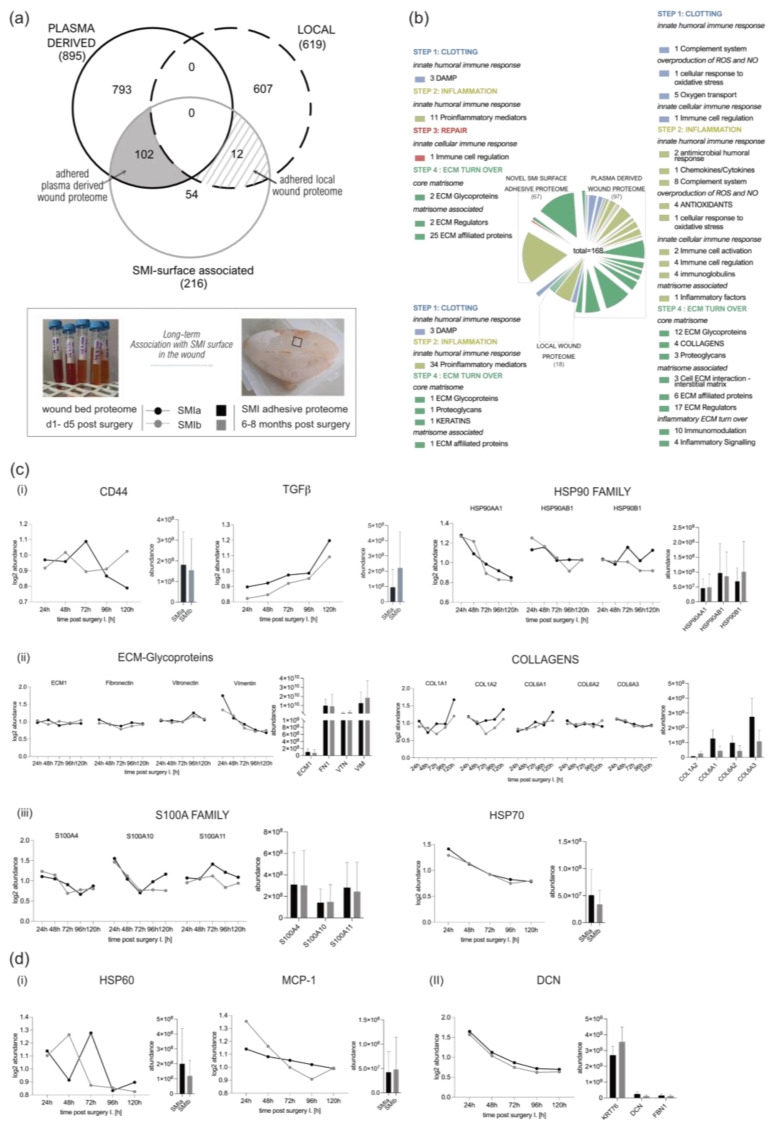
From wound to fibrosis. Comparative analysis of SMI-associated proteome 8 months post-op recruited and adhered from wound proteome 1–5 days post-SMI implantation. (**a**) Venn diagram showing the distribution of plasma-derived and local acute wound proteins associated at silicone tissue expander surface (*SMIa*: 4 µM and *SMIb* 60 µM surface roughness) 6–8 months post-implantation. (**b**) Functional categorization of SMI-associated wound proteome in the context of wound healing/tissue repair of common origin. (**c**,**d**) Comparative analysis of protein abundance in wound bed fluid 1–5 days post-implantation (**Left**; dots with connecting lines, median shown) and associated on SMI surface 6–8 months post-op (**Right**; bars, shown mean ± SD,): (**c**) plasma-derived (**i**) inflammatory response, (**ii**) ECM turn over, and (**iii**) profibrotic mediation, as well as (**d**) (**i**) local proinflammatory monocyte attraction and (**II**) ECM-turn over.

**Figure 8 biomolecules-13-00305-f008:**
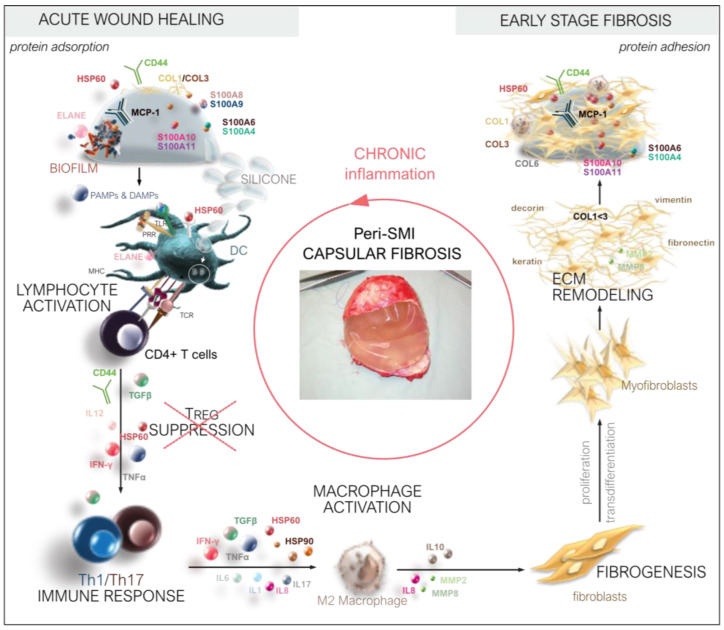
From wound to fibrosis. Chronological orchestration of immediate towards chronic FBR.

**Table 1 biomolecules-13-00305-t001:** Inclusion and exclusion criteria for the expander-immunology trial.

Inclusion Criteria	Exclusion Criteria
Female sex	Sever coagulation disorder, representing a potential contraindication for the elective surgery
Age >18 years	Rheumatic disease accompanied by oblkigatory intake of immunomodulating therapeutic agents
High-risk family history for breast and/or ovarian cancer and/or BRCA1/2 gene mutation carrie	Severe renal functional disorder: renal insufficiency status IV or V (estimated glomerulary filtration rate (GFR) <30 mL/min)
Planned bilateral mastectomy with simultaneous breast reconstruction	Active hematological or oncological disease
Signed Informed consent form	HIV-Infection
	Hepatitis-Infection
	Pregnancy or breast-feeding
	Intake of anti-inflammatory drugs
	Carrier of silicone implants (e.g., gastric banding, mammary implants)
	Subject is currently participating or intends to participate in another clinical trial that may interfere with the protocol of this study.
	Participants who have implanted devices that could be affected by a magnetic field (e.g., pacemakers, drug infusion devices, artificial sensing devices). When there is an alteration in hematologic and serum protein reference values post-chemotherapy.
	When there is a residual malignancy in the intended expansion site.
	Existing tissue at the intended expansion site is not adequate according to the surgeon’s criteria, because of previous radiation therapy, ulcerations, vascular compromise, history of compromised wound healing, or scar deformity.
	Radiation therapy before or after the expander placement can be associated with a higher rate of complications during the expansion and final implantation phases of the reconstructive process.
	Abscess or infection in the body in general.
	Participants with autoimmune diseases (e.g., lupus, scleroderma) or whose immune system is compromised (e.g., currently receiving immunosuppressive therapy such as steroids).
	Unsuitable tissue due to radiation damage on the chest wall, tight thoracic skin grafts or radical resection of the pectoralis major muscle.

**Table 2 biomolecules-13-00305-t002:** Intraindividual statistical comparison of analytical groups.

	SmoothSilk^®^		Mentor CPX4		
Surface Roughness	Ra ~ 4 µM		Ra ~ 60 µM		
	Mean	(±std)	Mean	(±std)	*p* value
age (y)	35.2	11.4	35.2	11.4	intraindividual comparison >0.9999
weight (kg)	71.4	24.5	71.4	24.5	
size (cm)	168.6	10.5	168.6	10.5	
BMI	25.1	6.7	25.1	6.7	
Bilateral prophylactic NSME resection weight [g]					
left breast	434.9	404.0	436.9	454.0	0.993196
right breast	334.2	257.5	337.9	174.4	0.975407
Prepectoral reconstruction volume [cc]					
left breast	405.5	156.3	392.6	151.8	0.877595
right breast	360.8	151.1	352.7	150.0	0.920737
intaoperative filling	254.7	169.4	254.7	169.4	>0.9999

**Table 3 biomolecules-13-00305-t003:** STRING kmeans (k = 4) cluster description of plasma-derived wound proteome.

	Cluster 1	Cluster 2	Cluster 3	Cluster 4
	RED	GREEN	YELLOW	BLUE
	CARBON METABOLISM in fibrolytic switches	MATRISOME	INFLAMMATORY IMMUNE CELL RESPONSE	INFLAMMSOME
proteins	281	120	159	308
nodes	281	120	159	308
edges	3592	503	1310	4526
average node degree	25.6	8.38	16.5	29.4
local clustering coefficient	0.51	0.52	0.54	0.49
expected number of edges	1284	18	163	479
PPI enrichment *p*-value				
	<1.0 × 10^−16^	<1.0 × 10^−16^	<1.0 × 10^−16^	<1.0 × 10^−16^

**Table 4 biomolecules-13-00305-t004:** Numbers of inflammatory matrisome proteins in plasma-derived wound proteome after silicon device implantation.

	Inflammatory Matrisome Class	Number of Common Plasma Derived Wound Proteins	Number of Differentially Expressed Plasma Derived Wound Proteins	Ratio [%]
Innate humoral immune response	Antimicrobial humoral response	46	15	33%
	Chemokines/cytokines	16	5	31%
	Complement system	43	9	21%
Innate cellular immune response	Immune cell activation	15	5	33%
	Immune cell regulation	11	8	73%
	Immunoglobulins	30	0	0%
Overproduction of ROS and NO	Cellular response to oxidative stress	47	24	51%
	Oxygen transport	5	5	100%
	Antioxidants	5	5	100%
Mechanical stress	Muscle component	4	6	150%
	Hormone	1	1	100%
Core matrisome	ECM glycoproteins	59	13	22%
	Collagens	8	1	13%
	Proteoglycans	4	0	0%
Matrisome associated	ECM regulators	87	12	14%
	ECM affiliated proteins	23	6	26%
	Cell ECM interaction—interstitial matrix	5	2	40%
	Secreted factors	16	9	56%
	Inflammatory factors	6	2	33%
	Immunomodulation	3	0	0%
	Fibroblast migration	1	1	100%

More details can be found in Appendix A.

**Table 5 biomolecules-13-00305-t005:** STRING kmeans cluster description of local wound proteome.

	Cluster 1	Cluster 2	Cluster 3	Cluster 4
	RED	GREEN	YELLOW	BLUE
	CARBON METABOLISM in fibrolytic switches	MATRISOME	INFLAMMATORY IMMUNE CELL RESPONSE	INFLAMMSOME
proteins	150	221	117	123
nodes	150	221	117	123
edges	306	967	575	967
average node degree	4.08	0.17	9.83	15.7
local clustering coefficient	0.394	0.38	0.531	0.515
expected number of edges	49	307	123	242
PPI enrichment *p*-value				
	<1.0 × 10^−16^	<1.0 × 10^−16^	<1.0 × 10^−16^	<1.0 × 10^−16^

**Table 6 biomolecules-13-00305-t006:** Number of inflammatory matrisome proteins in local wound proteome after silicon device implantation.

	Inflammatory Matrisome Class	Number of Common Local Wound Proteins	Number of Differentially Expressed Local Wound Proteins	Ratio [%]
Innate humoral immune response	DAMP	12	0	0%
	PAMP	1	0	0%
	Antimicrobial humoral response	1	1	100%
	Chemokines/Cytokines	1	0	0%
	Complement system	13	2	15%
	Blood clotting pathway	6	3	50%
	Proinflammatory mediators	179	36	20%
	Inflammation-resolving	2	0	0%
Innate humoral immune response	Innate Immune Gene Expression	1	0	0%
	Immune cell activation	2	0	0%
	Immune cell regulation	11	3	27%
	Immunoglobulins	20	0	0%
	Cellular response to oxidative stress	2	0	0%
	Fibrosis mediator	1	0	0%
Core matrisome	ECM Glycoproteins	15	1	7%
	Keratins	6	2	33%
	Collagens	4	0	0%
	Proteoglycans	5	1	20%
Matrisome associated	ECM affiliated proteins	13	5	38%
	ECM Regulators	16	1	6%
	ECM affiliated proteins	11	0	0%
	Cell ECM interaction—interstitial matrix	4	1	25%
	Secreted factors	2	2	100%
	Inflammatory Signalling	1	0	0%
Oncologic marker	Mammaglobin A	1	1	100%

More details can be found in Appendix A.

**Table 7 biomolecules-13-00305-t007:** Acute wound proteome association with SMI surfaces 6–8 months post-implantation.

	Inflammatory Matrisome Class	Number of Plasma Derived Wound Proteins d1–d5 Post SMI Implantation	Number of Plasma Derived Wound Proteins Associated with SMI Surface 6–8 Months Post SMI Implantation	Ratio [%]	Number of Local Wound Proteins	Number of Local Wound Proteins Associated with SMI Surface 6–8 Months Post SMI Implantation	Ratio [%]
Innate humoral immune response	DAMPs				1	1	100%
	Antimicrobial humoral response	46	3	7%			
	Chemokines/Cytokines	16	1	6%			
	Proinflammatory mediation (ribosomal proteins)				179	8	4%
	Complement system	43	11	26%			
Innate cellular immune response	Immune cell activation	15	2	13%			
	Immune cell regulation	11	6	55%			
	Immunoglobulins	30	5	17%			
Overproduction of ROS and NO	Cellular response to oxidative stress	47	4	9%			
	Oxygen transport	5	9	180%			
	Antioxidants	5	4	80%			
Mechanical stress	Muscle component	4	1	25%			
	Hormone	1	0	0%			
Inflammatory response		223	46	21%	180	9	5%
Core matrisome	ECM glycoproteins	59	13	22%	15	2	13%
	Collagens	8	5	63%	4	0	0%
	Proteoglycans	4	3	75%	5	1	20%
	Keratins				6	0	0%
Matrisome associated	ECM regulators	87	15	17%			
	ECM affiliated proteins	23	7	30%			
	Cell ECM interaction—interstitial matrix	5	2	40%			
	Secreted factors	16	11	69%			
	Inflammatory factors	6	1	17%			
	Immunomodulation	3	3	100%			
	Fibroblast migration	1	0	0%			
ECM Turn-Over		212	60	28%	30	3	10%

## Data Availability

The mass spectrometry proteomics data have been deposited to the ProteomeXchange Consortium (http://proteomecentral.proteomexchange.org, accessed on 3 February 2023) via the PRIDE partner repository with the dataset identifier PXD039840 and are publicly available as of the date of publication. The data presented in this study are available on request from the corresponding author due to data privacy protection. The trial basic summary results have been deposited within ClinicalTrials.gov (ID NCT05648929) registry and are publicly available as of the date of publication.

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
