# Peer review of "Quantitative Proteomic Characterization of Foreign Body Response towards Silicone Breast Implants Identifies Chronological Disease-Relevant Biomarker Dynamics"

_biomolecules, 2023, doi:10.3390/biom13020305_

Round 1
Reviewer 1 Report
Summary
The manuscript submitted by Ines Schoberleitner et al. is aimed to discover potential triggers or biomarkers responsible of exaggerated fibrous capsule formation around silicone mammary implant (SMI). To this end, the authors refer to proteomics, spectrometry and chromatography analysis. In order to study this foreign body response, a monocentric, randomized, double-blind controlled clinical study was conducted. A total of 7 female patients who were undergoing simultaneous prophylactic bilateral nipple-sparing mastectomy (NSME) and tissue expander-based breast reconstruction were taken into account. The proteomes of preoperative plasma (-d1), wound fluid derived from surgical drainages following NSME (d1-d5) and tissue adhered the SMI surface 8 months postoperatively were investigated. Two types of breast implants were used and randomly attributed. The authors concluded that this study offers valuable insights over the composition of the wound proteome, comprised of a systemic plasma-derived and a local (expressed in local tissue) wound proteome, the first serving as an inflammatory foreign-body responder, and the second as a proinflammatory mediator. Also, the authors revealed potential early markers, such as proteins S100A8/A9, and potential long-term markers, including COL1, HSP90, S100A4, and ELANE with high diagnostic sensitivity.
Title and abstract: The title and abstract are appropriate for the contents of the text.
Introduction: The authors have adequately interpreted and presented the data currently available in the literature, in addition to discuss the shortcomings of the proposed ideas. The aim of the study is well stated and sustained by scientific data.
Materials and methods: The methodology for patient inclusion and exclusion was presented clearly. The protocol regarding data collection, follow-up, collection of biological samples and laboratory analysis is well established and structured. The statistical analysis is described in detail.
Results: The information presented is nicely supported by figures and tables. The tables, images and figures are of good quality and well interpreted. Although the authors presented their findings in a comprehensive and structured manner, this chapter includes numerous paragraphs and rich information that should be included only in Materials and methods section. This section should be revised and should encompass only the data obtained after laboratory and statistical analysis.
Discussions: The results are discussed in relation to the evidence currently available in the literature. The limitations and strengths of the present study are adequately presented.
Conclusions: The conclusions of the authors are appropriately cautious given the limitations of the study.
Lastly, the manuscript is presented in a structured manner with the advice of some revision. Most of the cited references are recent and relevant, but there are some self citations. The use of language is adequate.
Reviewer 2 Report
Schoberleitner et al. performed proteomics analysis to study the molecular changes induced by silicon mammary implant (SMI) insertion, recapitulating immediate tissue damage response in the acute wound proteome as well as adsorption of chronic inflammatory proteins at implant surface. The authors analysed samples from the wound formed around SMI five days post-implantation as well as depleted plasma samples from 7 women. The authors also investigated the long-term adsorbed acute wound and the fibrosis-associated proteomes.
The study is interesting and potentially publishable however the manuscript and particularly the experimental design is written in a quite confusing way, and it is not easy for the reader to follow. The entire quantification data should be supplied in the supplementary tables, before or after normalization and/or any data transformation. The supplementary excel file seems corrupted and unfortunately I was unable to fully review it. A reference sample was used across the different TMT sets but it is not clear in the manuscript how data was normalized or scaled and whether there were batch effects that needed correction.
I also have major comments about the data analysis performed and presentation of results in this study.
Some examples:
Figure S1. Panels i and ii do not show correlation plots but box plots. The figure also has confusing numbering of panels. It is not clear what panels iii show as there are no axis labels. The abundance values in fig S1 should be log10 or log2 transformed. In the main text the authors state: “We obtained a high coefficient of correlation (R2) value for the sample TMT abundance ratios compared to the standard” however no ratios are shown in fig S1.
Figure 3b. “PCA was obtained with a median log2 abundance of the differential proteins found in all samples.” It is fundamentally wrong to use differentially regulated proteins after statistical filtering to perform PCA as the separation is “forced” and does not represent the unbiased classification of the samples. Panel c is not a scatter plot. What is the conclusion drawn from plot d?
Figure S2b is not a PCA plot and it doesn’t show protein abundance ratio either. The panel numbering and legend in fig S2 is confusing too (a, b, etc with subdivision i,ii, etc). The “high” correlation in panels i is not obvious at all.
The manuscript needs some editing throughout:
For example:
TMT is not “absolute quantitation” as mentioned in the abstract, but relative quantification.
“3.4. Quantification and data integrity of intraindividual comparative proteomic profiling in 405 plasma, wound, and SMI-adhesive proteome”. What does data integrity mean here?
“However, the method of sample collection used is an integral step in the research process”. The meaning is not clear.
Ethics statement and Patient characteristics sections are not results.
The paragraph in the discussion: “There is a tendency of autoimunologists to forbid SMIs altogether, due to,e.g. the high percentage of SMI carriers presenting with circulating proteins - antibodies against G protein-coupled receptors (GPCRs) in small autonomic nerve fibers[58].” seems misplaced.
Several words are underlined or with bold font.
Round 2
Reviewer 1 Report
The current form of the manuscript complied with the recommendations and is appropriate to be published.
Reviewer 2 Report
The authors have addressed my comments and the manuscript is suitable for publication.